# Functional and structural deficiencies of Gemin5 variants associated with neurological disorders

Rosario Francisco-Velilla[1], Azman Embarc-Buh[1], Francisco del Caño-Ochoa[2,3], Salvador Abellan[1], Marçal Vilar[2], Sara Alvarez[4], Alberto Fernandez-Jaen[5,6], Sukhleen Kour[7], Deepa S Rajan[7], Udai Bhan Pandey[7], Santiago Ramón-Maiques[2,3], Encarnacion Martinez-Salas[1]

**Dysfunction of RNA-binding proteins is often linked to a wide range of human disease, particularly with neurological conditions. Gemin5 is a member of the survival of the motor neurons (SMN) complex, a ribosome-binding protein and a translation reprogramming factor. Recently, pathogenic mutations in *Gemin5* have been reported, but the functional consequences of these variants remain elusive. Here, we report functional and structural deficiencies associated with compound heterozygosity variants within the *Gemin5* gene found in patients with neurodevelopmental disorders. These clinical variants are located in key domains of Gemin5, the tetratricopeptide repeat (TPR)–like dimerization module and the noncanonical RNA-binding site 1 (RBS1). We show that the TPR-like variants disrupt protein dimerization, whereas the RBS1 variant confers protein instability. All mutants are defective in the interaction with protein networks involved in translation and RNA-driven pathways. Importantly, the TPR-like variants fail to associate with native ribosomes, hampering its involvement in translation control and establishing a functional difference with the wild-type protein. Our study provides insights into the molecular basis of disease associated with malfunction of the Gemin5 protein.**

## Introduction

RNA-binding proteins (RBPs) perform critical roles in RNA metabolism, regulating all steps of gene expression (Gehring et al, 2017; Diaz–Muñoz & Osma-Garcia, 2021). The modular organization of RBPs, generally consisting of RNA-binding domains and protein–protein interaction modules (Lunde et al, 2007), provides multiple activities to these factors. Indeed, malfunction of distinct RBPs has been related to human diseases (Gebauer et al, 2021; Smith & Costa,

2021). On the other hand, dysregulation of the binding activity of a given protein can produce widespread effects on multiple RNA-dependent processes and, as a result, challenge the identification of the molecular mechanism contributing to disease.

Gemin5 is predominantly a cytoplasmic protein first described as a component of the survival of motor neurons (SMN) complex (Matera et al, 2019). In humans, the SMN complex comprises nine members (SMN, Gemins2–8, and unr-interacting protein [Unrip]) (Otter et al, 2007). The SMN complex plays a critical role in the biogenesis of small nuclear RNPs (Pellizzoni et al, 2002; Lau et al, 2009), the components of the splicing machinery (Kastner et al, 2019). Altered levels of the SMN protein causing defects in the SMN complex assembly lead to spinal muscular atrophy, a severe to mild form of disease depending upon the SMN protein levels (Burghes & Beattie, 2009). In addition, the SMN protein affects other processes such as translation regulation (Sanchez et al, 2013; Lauria et al, 2020), muscle architecture (Rajendra et al, 2007), or transport and assembly of microRNPs or telomerase RNPs (Mourelatos et al, 2002).

Gemin5 was initially identified as the protein responsible for the recognition and delivery of small nuclear RNAs to snRNPs (Battle et al, 2006; Yong et al, 2010). Nonetheless, a large fraction of the Gemin5 protein is found in the cytoplasm outside of the SMN complex (Battle et al, 2007), strongly suggesting that the protein may contribute to additional cellular processes. Accordingly, Gemin5 acts as a hub for several networks performing diverse key cellular functions. This multifunctional protein has been shown to act as a regulator of translation (Pacheco et al, 2009; Workman et al, 2015; Francisco-Velilla et al, 2018), as a ribosome-interacting protein (Francisco-Velilla et al, 2016; Simsek et al, 2017), as a reprogramming factor in zebrafish lateral line hair cells (Pei et al, 2020), as a signal recognition particle-interacting protein (Piazzon et al, 2013), and as a trans-splicing factor (Philippe et al, 2017). In addition, Gemin5 has been identified as a member of RNP networks associated to distinct cytoplasmic aggregates (Jiang et al, 2018; Vu et al, 2021;

[1]Centro de Biología Molecular Severo Ochoa (CBMSO), Consejo Superior de Investigaciones Científicas - Universidad Autónoma de Madrid (CSIC-UAM), Madrid, Spain [2]Instituto de Biomedicina de Valencia (IBV-CSIC), Valencia, Spain [3]Centro de Investigación Biomédica en Red de Enfermedades Raras (CIBERER), Instituto de Salud Carlos III (ISCIII), Madrid, Spain [4]New Integrated Medical Genetics (NIMGENETICS), Madrid, Spain [5]Neuropediatric Department, Hospital Universitario Quirónsalud, Madrid, Spain [6]School of Medicine, Universidad Europea de Madrid, Madrid, Spain [7]Department of Pediatrics, Children's Hospital of Pittsburgh, University of Pittsburgh Medical Center, Pittsburgh, PA, USA

Correspondence: emartinez@cbm.csic.es

Wollen et al, 2021). More importantly, *Gemin5* variants were recently linked with human neurodevelopmental disorders, perturbing distinct pathways as compared with defects in the SMN protein (Kour et al, 2021; Saida et al, 2021; Rajan et al, 2022). However, the molecular basis of Gemin5 dysfunction remains elusive.

The human Gemin5 protein is organized in functional domains with a distinctive structural organization (Fig 1A). The N-terminal part contains two juxtaposed seven-bladed WD40 domains (Jin et al, 2016) that recognize the Sm-site of snRNAs and the cap via base-specific interactions (Tang et al, 2016; Xu et al, 2016). The crystal structure of the central region revealed a tetratricopeptide repeat (TPR)–like domain with 17 $\alpha$-helices that oligomerizes as a canoe-shaped homodimer (Moreno-Morcillo et al, 2020). Insertion of a single-point substitution (A951E) at the closest position between the two subunits disrupts the assembly of the dimer. In addition, the mutation A951E confers a strong decrease in the Gemin5 interactome and results in the loss of factors connected to RNA processing, translation regulation, and spliceosome assembly, among others. Taken together, these results strongly suggest that dimerization (and perhaps, multimerization) is an evolutionary preserved trait of Gemin5.

The most C-terminal part of Gemin5 harbors a noncanonical RNA-binding site (RBS) comprising two domains, designated as RBS1 and RBS2 (Fernandez-Chamorro et al, 2014). The RBS1 moiety recognizes an internal region of *Gemin5* mRNA, stimulating its own translation and counteracting the negative effect of Gemin5 in global protein synthesis (Francisco-Velilla et al, 2018). Structural analysis of the RBS1 polypeptide shows the presence of an un-folded region (Embarc-Buh et al, 2021), a feature frequently found in intrinsically disordered regions of proteins characterized by having multiple interactors, either RNA or proteins (Järvelin et al, 2016).

Here, we sought to establish the impact of *Gemin5* missense variants found in compound heterozygosity in patients developing neurodevelopmental disorders. Importantly, these variants are located in two separate domains of the protein, the TPR-like dimerization module and the noncanonical RBS1 domain, which are colinear on the G5$_{845-1508}$ region of the protein (Fig 1A). Considering the implications of these functional domains for the multiple activities of Gemin5 (Francisco-Velilla et al, 2019), we took advantage of the known properties of the G5$_{845-1508}$ fragment to analyze the impact of these clinical variants in Gemin5 protein dimerization, ribosome association, protein–protein interaction, and translation control.

We found that individual substitutions within the dimerization domain and the noncanonical RBS domain of Gemin5 lead to protein malfunction. Two nearly placed variants within the TPR-like domain impair the dimerization ability of Gemin5 protein in living cells, whereas the variant on the RBS1 domain decreases Gemin5 protein stability. Mass spectrometry analysis revealed that these variants abrogate the association of cellular proteins involved in RNA splicing and translation, among other cellular processes. Consistent with the reduced capacity to interact with factors involved in translation, the proteins harboring substitutions on the TPR-like domain elicit a reduced capacity to sediment with native ribosomes. Furthermore, in contrast to the enhancing activity of the wild-type (WT) G5$_{845-1508}$ protein on cap-dependent and selective translation, the mutants analyzed in this study failed to do so in human cells. Together, our results provide molecular insights to understand the implication of *Gemin5* variants in disease.

# Results

### Gemin5 variants on conserved residues of the TPR-like moiety and the RBS1 domain lead to disease

The exome sequencing of three patients with neurological disorders revealed the presence of biallelic variants within *Gemin5*, absent in the normal population (GnomAD database). Most interesting, the substitutions found in these patients are present in compound heterozygosity (Fig 1B). Index case 1 is one individual with a splicing variant on intron 4 in one allele (frequency 60%) that is predicted to yield a truncated protein, and a missense variant in exon 22 in the other allele (frequency 50%) producing the R1016C substitution within the TPR-like domain (Table 1 and Fig 1A). This individual is a female with intellectual disability, autism disorder, delay staturo-ponderal growth, microcephaly, and mild dysmorphic features but no motor problems. Both parents and a sister with monoallelic *Gemin5* variants are healthy, reinforcing the crucial impact of compound heterozygosity in disease. A second case corresponds to a family where two siblings (patients 2 and 3 in Table 1) with severe to mild neurological disorders carry a D1019E missense substitution within the TPR-like moiety in one allele and a L1367P substitution within the RBS1 region in the other allele (Fig 1B). Individuals 2 and 3 are female and male, respectively, with ataxia, hypotonia, developmental delay, cerebellar atrophy, motor delay, and cognitive delay (Kour et al, 2021).

The alignment of Gemin5 protein sequences obtained from vertebrate species, including mammals, birds, amphibians, reptiles, and fishes (Fig S1) shows that D1019 is fully conserved, whereas R1016 shows substitution to His, Leu, and Glu but never to Cys. L1367 locates within a predicted helix of RBS1, with substitutions to Val, Phe, and Cys but not to Pro that is a helix-breaker amino acid. Therefore, each of these clinical variants affect conserved residues of the protein which is likely to affect the physiological functions.

### Effect of the R1016C and D1019E mutations on the structural conformation of the TPR-like module

Understanding the involvement of the Gemin5 domains in the different activities of this protein is key to interpret malfunction of the clinical variants. Disruption of the dimerization ability interferes with critical functions of this protein (Moreno-Morcillo et al, 2020). Therefore, we sought to analyze the impact of the pathogenic variants R1016C and D1019E on the Gemin5 TPR-like dimerization module. This region forms a canoe-shaped homodimer with each subunit folding in an extended palisade of $\alpha$-helices. Residues R1016 and D1019 located in a 6-aa loop between helices 12 and 13 (loop $\alpha$12-13) and their side chains are involved in a network of electrostatic interactions (Fig 1C). R1016 makes a salt bridge with E1018, an H-bond through the carbonyl oxygen with the amide nitrogen of D1019, and a water-mediated H-bond between the amide N and the side chain of D1019. Thus, replacing R1016 by a Cys could cause the loss of the salt bridge and a distortion of the solvent-exposed loop because of the increased hydrophobicity of this residue (Fig 1C). On the other hand, the side chain of D1019 makes an H-bond with the amide group of V1021, stabilizing the

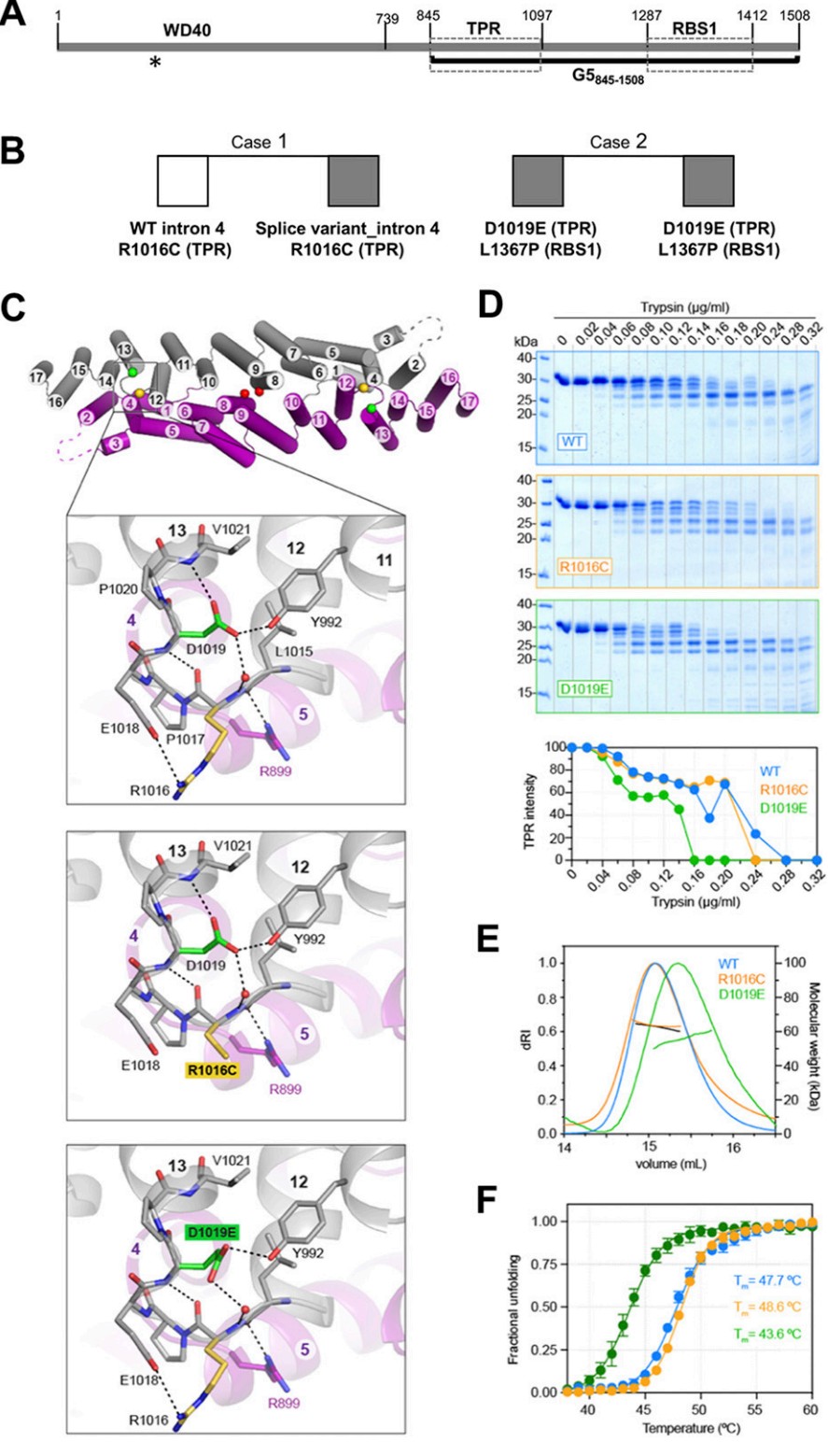

**Figure 1. Gemin5 protein organization, pathogenic mutants, and structural properties of the variants on the TPR-like domain.**
**(A)** Schematic of functional domains of Gemin5. The regions encompassing the WD40 domain, the TPR-like dimerization module, and the noncanonical RBS1 domain are plotted. The position of amino acids flanking each domain is indicated at the top. A thick black line depicts the region included in the protein G5$_{845-1508}$. An asterisk marks the last residue of the truncated protein (about 220 residues) predicted by the splice variant mutation. **(B)** Summary of *Gemin5* biallelic variants found in the exome sequencing of patients developing neurodevelopmental disorders. **(C)** Cartoon representation of Gemin5 TPR-like homodimer with each subunit shown in different color. The numbers for the helices are indicated. Dashed lines indicate the flexibly disordered loop α2-3. The spheres indicate the position of mutated residues. The enlarged area shows a zoom view of the loop α12-13 with residue R1016 (yellow) and D1019 (green). Dashed lines indicate electrostatic interactions. Models of the loop α12-13 bearing mutations R1016C or D1019E are shown below. **(D)** Limited protease digestion of the WT and mutant proteins. Equal amounts of purified Gemin5 TPR-like proteins (WT, R1016C, and D1019E) were treated with serial dilutions of trypsin (0–0.32 μg/ml) and subjected to SDS–PAGE. A representative example of a Coomassie blue–stained gel is shown. Quantification of the 30-kD protein band shows differences in the sensitivity to trypsin cleavage. **(E, F)** Analysis of the WT, R1016C, and D1019E proteins by SEC-MALS (E) or differential scanning fluorimetry (F). Values are represented as the mean ± SD from six independent measurements for each protein.

N-terminal dipole of helix 13 and an H-bond with the phenolic oxygen of Y992 in the helix 11. A glutamate replacing D1019 would clash with neighboring side chains, and only one permissive position would maintain the interactions with Y992 and with the water molecule but would not be adequate for the capping of helix 13 (Fig 1C).

To interrogate the impact of the pathogenic variants on the structure of the protein, we produced the TPR-like dimerization

**Table 1. Patients' whole-exome sequencing[a,b,c].**

| Chr. | Gene | Intron/exon | Nucleotide change | Amino acid change | Mutation type |
|------|------|-------------|-------------------|-------------------|---------------|
| 5 | Gemin5 | 4 | NM_015465.4: c.662-2A>G | - | 5′splice site |
| 5 | Gemin5 | 22 | NM_015465.4: c.3046C>T | p.Arg1016Cys | Missense |
| 5 | Gemin5 | 22 | NM_015465.5: c.3057C>A | p.Asp1019Glu | Missense |
| 5 | Gemin5 | 26 | NM_015465.5: c.41000T>C | p.L1367P | Missense |
| 5 | Gemin5 | 22 | NM_015465.5: c.3057C>A | p.Asp1019Glu | Missense |
| 5 | Gemin5 | 26 | NM_015465.5: c.41000T>C | p.L1367P | Missense |

[a]Female, compound heterozygosity, intellectual disability, autism disorder, delay staturo-ponderal growth, microcephaly, mild dysmorphic features, and no motor problems.
[b]Female, compound heterozygosity, ataxia, hypotonia, developmental delay, severe cerebellar atrophy, motor delay, mild cognitive delay, slowed visual pursuit, and mild oculomotor apraxia (Kour et al, 2021).
[c]Male, compound heterozygosity, ataxia, hypotonia, moderate cerebellar atrophy, motor delay, tremor, hyperreflexia, and mild dysarthria (Kour et al, 2021).

domain bearing R1016C or D1019E substitution following the same expression and purification protocols as for the WT protein. The mutants were purified at a lower yield than the WT protein and required higher salt concentration (from 50 to 200 mM) for their solubility. Limited proteolysis analysis showed that D1019E was more susceptible to trypsin digestion than the R1016C and the WT protein (Fig 1D). However, both mutants formed stable homodimers at 200 mM salt concentration, as observed by SEC-MALS, although D1019E showed signs of instability (Fig 1E). In fact, we measured by differential scanning fluorimetry (Thermofluor) that the melting temperature (Tm; midpoint of the unfolding transition) of D1019E was 4°C lower than that of R1016C or the WT protein (Fig 1F). These results suggest a greater flexibility of the TPR moiety carrying the D1019E substitution, whereas the effect of the R1016C variant was not apparent under these conditions. Taken together, the structural changes induced by mutations on the TPR-like domain led us to investigate their potential effect on Gemin5 functions.

### Gemin5 variants in the TPR-like domain decrease protein dimerization

Conserved residues in the TPR-like region are involved in interactions within the subunit and across the dimer interface (Moreno-Morcillo et al, 2020). This moiety (residues 845–1,097) drives dimerization of the purified polypeptide in vitro and in human cells when present within the G5$_{845-1508}$ fragment, which encompasses both the TPR-like and the RBS1 domains (Fig 2A). Hence, we asked whether the clinical variants R1016C and D1019E in the TPR-like module would interfere with the dimerization capacity of the protein in the cell, which could ultimately impact the functions of this multitasking protein.

To determine the recruitment of full-length Gemin5, we expressed the variants in the G5$_{845-1508}$ tandem affinity purification (TAP) context (Fig 2A). The dimerization defective mutant A951E was used as a control (Moreno-Morcillo et al, 2020). We also generated the L1367P construct for completeness because this mutation is present in a different domain of the protein. Variants R1016C and D1019E were expressed in HEK293 cells to a similar extent than the WT protein and the dimerization mutant A951E, whereas the mutant L1367P rendered lower levels of the protein (Fig 2A). Then, the intensity of coimmunoprecipitation (co-IP) of the full-length

Gemin5 protein was measured in protein complexes purified by TAP (Fig S2A) using G5$_{845-1508}$-WT and the mutants A951E, R1016C, D1019E, or L1367P as baits. Gemin5 and G5$_{845-1508}$ were immuno-detected using anti-Gemin5 (Fig 2B, top panel). In each case, the intensity of recruited Gemin5 was normalized to the intensity of purified G5$_{845-1508}$. The recruitment of Gemin5 by G5$_{845-1508}$-WT was 10-fold higher than that observed by the dimerization mutant A951E. Interestingly, the substitutions R1016C and D1019E impaired co-IP of the endogenous Gemin5 protein, reducing the intensity to 58% and 62% relative to the WT protein, respectively. In comparison, the construct L1367P did not have a significant effect on Gemin5 co-IP (Fig 2B, bottom panel).

Next, we identified by mass spectrometry the Gemin5 protein copurifying with G5$_{845-1508}$ variants expressed in HEK293 cells (Fig 2C). After TAP purification and LC–MS/MS, PEAKS analysis of Gemin5 amino acid sequences was used to identify the presence of trypsin fragments corresponding to unique peptides of the N-terminal (1–844) and the C-terminal parts (845–1,508) of Gemin5 (Fig S2B). Then, total reads of the unique peptides within positions 1–844 and 845–1,508 of Gemin5 were accounted. Relative to G5$_{845-1508}$-WT, the ratio of peptide reads belonging to the N-terminal relative to the C-terminal region was diminished in R1016C and D1019E (0.77 and 0.56, respectively), revealing a reduced capacity to recruit Gemin5 (Fig 2C). In agreement with the co-IP data (Fig 2B), the dimerization mutant A951E was severely defective in Gemin5 recruitment, whereas negligible decrease in Gemin5 recruitment was observed for the L1367P construct (0.90) (Fig 2C). The mutations R1016C and D1019E within the TPR-like domain decrease the G5$_{845-1508}$ re-cruitment ability of the Gemin5 protein, reflecting that individually these variants impair the formation of the dimer complex in a competitive cellular environment.

### The Gemin5 mutants display a reduced capacity to associate factors involved in translation control and RNA metabolism

The dimerization defects associated with the substitutions present in the TPR-like module of Gemin5 prompted us to study global effects on protein–protein interactions. To this end, we identified the cellular factors associated to each TAP-tagged G5$_{845-1508}$ variant (Fig S3A and B and Supplemental Data 1). Of note, the identified factors revealed a closer similarity among the three clinical

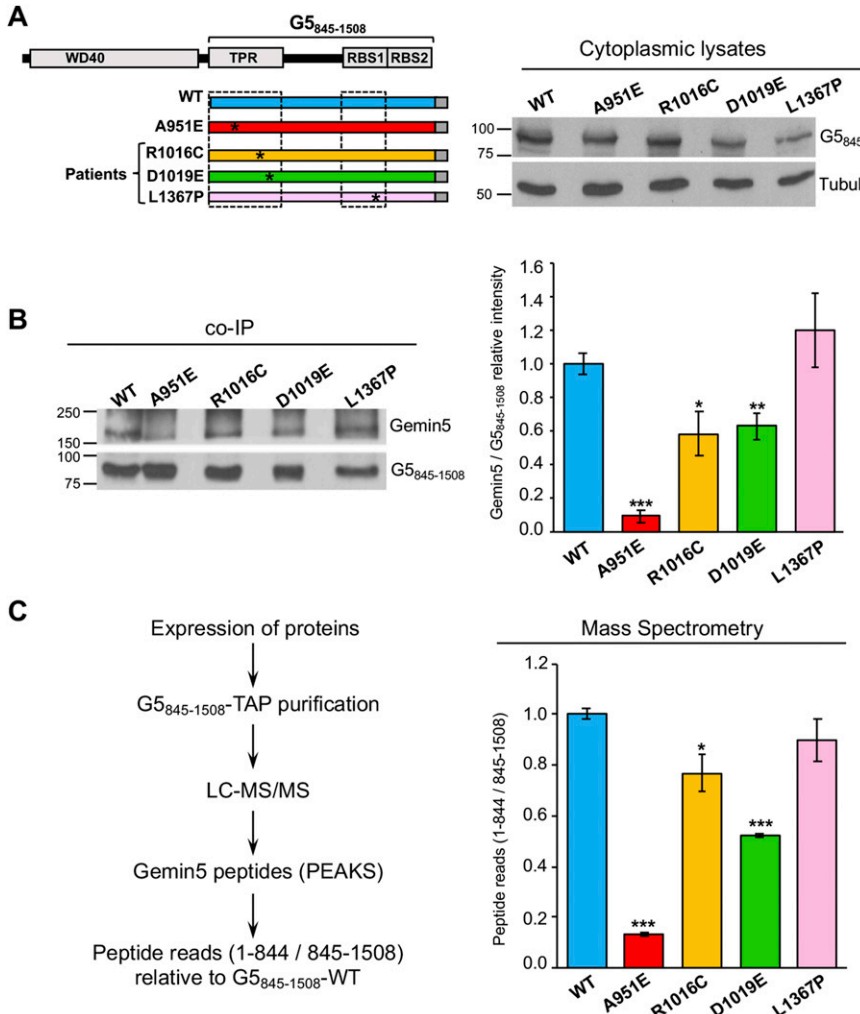

**Figure 2. Mutations in the TPR-like domain impair the capacity of G5$_{845-1508}$ to recruit the endogenous Gemin5 protein in living cells.**
**(A)** Expression of G5$_{845-1508}$-TAP proteins. HEK293 cells were transfected with plasmids expressing G5$_{845-1508}$-WT-TAP or G5$_{845-1508}$-A951E, R1016C, D1019E, or L1367P proteins. Twenty-four hours later, protein expression in cell lysates was followed by immunoblot using anti-CBP antibody. Tubulin was used as loading control. **(B)** Co-immunoprecipitation of Gemin5 with G5$_{845-1508}$-TAP purification samples obtained from soluble cell extracts expressing G5$_{845-1508}$-WT-TAP or the G5$_{845-1508}$ variants were detected by WB using the anti-Gemin5 antibody (left panel). The relative intensity Gemin5/G5$_{845-1508}$ (mean ± SEM) is represented (right panel). **(C)** The Gemin5 endogenous protein copurifying with G5$_{845-1508}$-WT or the G5$_{845-1508}$ mutants was identified by mass spectrometry (left panel). The number of reads of unique peptides corresponding to the N-terminal region of Gemin5 (residues 1–844) was made relative to those found in the G5$_{845-1508}$ region (845–1,508) (right panel). Asterisks denote *P*-values (*$P < 0.05$, **$P < 0.01$, ***$P < 0.001$). Source data are available for this figure.

mutations (37%) than with the WT protein (17%), providing evidence for common features among the mutants (Fig S3C).

Next, we sought to investigate the functional groups overrepresented within the factors copurifying with the proteins R1016C, D1019E, or L1367P relative to G5$_{845-1508}$-WT by Gene Ontology (GO) annotation of the biological processes. GO classification in functional categories using BiNGO (Cytoscape platform) showed a distribution in statistically significant overrepresented nodes (Maere et al, 2005). The top GO functional networks copurifying with G5$_{845-1508}$-WT are RNA-dependent processes, such as Translation ($2 \times 10^{-50}$), RNA metabolism ($8 \times 10^{-25}$), mRNA regulation ($2 \times 10^{-5}$), RNA localization ($2 \times 10^{-4}$), and SMN complex ($4 \times 10^{-4}$) (Fig 3A and B). In marked difference with the WT protein, the interaction networks related to translation and RNA metabolism were strongly reduced in the mutants R1016C and D1019E (Fig 3A, C, and D). We also noticed an important decrease in the number of factors associated with the RBS1 variant L1367P (Fig 3A and E). The mRNA regulation network only remained in D1019E, whereas RNA localization was below the threshold ($5 \times 10^{-3}$) in all the mutants. Loss of the SMN complex network was a marked feature of R1016C and D1019E mutants. In contrast, the L1367P mutant retained this network, although with a

slightly lower *P*-value ($3 \times 10^{-3}$) than the WT protein ($4 \times 10^{-4}$) and showed a smaller number of nodes (Fig 3A and E). We also noticed that the G5$_{845-1508}$-mutants copurified with members of the stress response (*P*-values ranging from $2 \times 10^{-3}$ to $9 \times 10^{-4}$).

The specific factors interacting with the WT and mutated variants are highlighted in Fig 3F. Most notably, the G5$_{845-1508}$-WT protein specifically associated a large number of ribosomal proteins, as well as numerous RBPs and members of the SMN complex. Most of these factors remained undetected in the variants analyzed in this study. We conclude that there is a crosstalk between the ability of G5$_{845-1508}$ to recruit the endogenous Gemin5 driven by the dimerization module and its capacity to associate partners of the SMN complex, as well as factors involved in translation and RNA metabolism.

## The Gemin5 RBS1 variant confers protein instability

The particular combination of recessive biallelic variants found in patients— in one case a splice variant in intron 4 with a dimerization mutant protein (R1016C), and in the other case a dimerization mutant protein (D1019E) with a point substitution in the RBS1 domain (L1367P) (Fig 1B) —could be compatible with altered levels

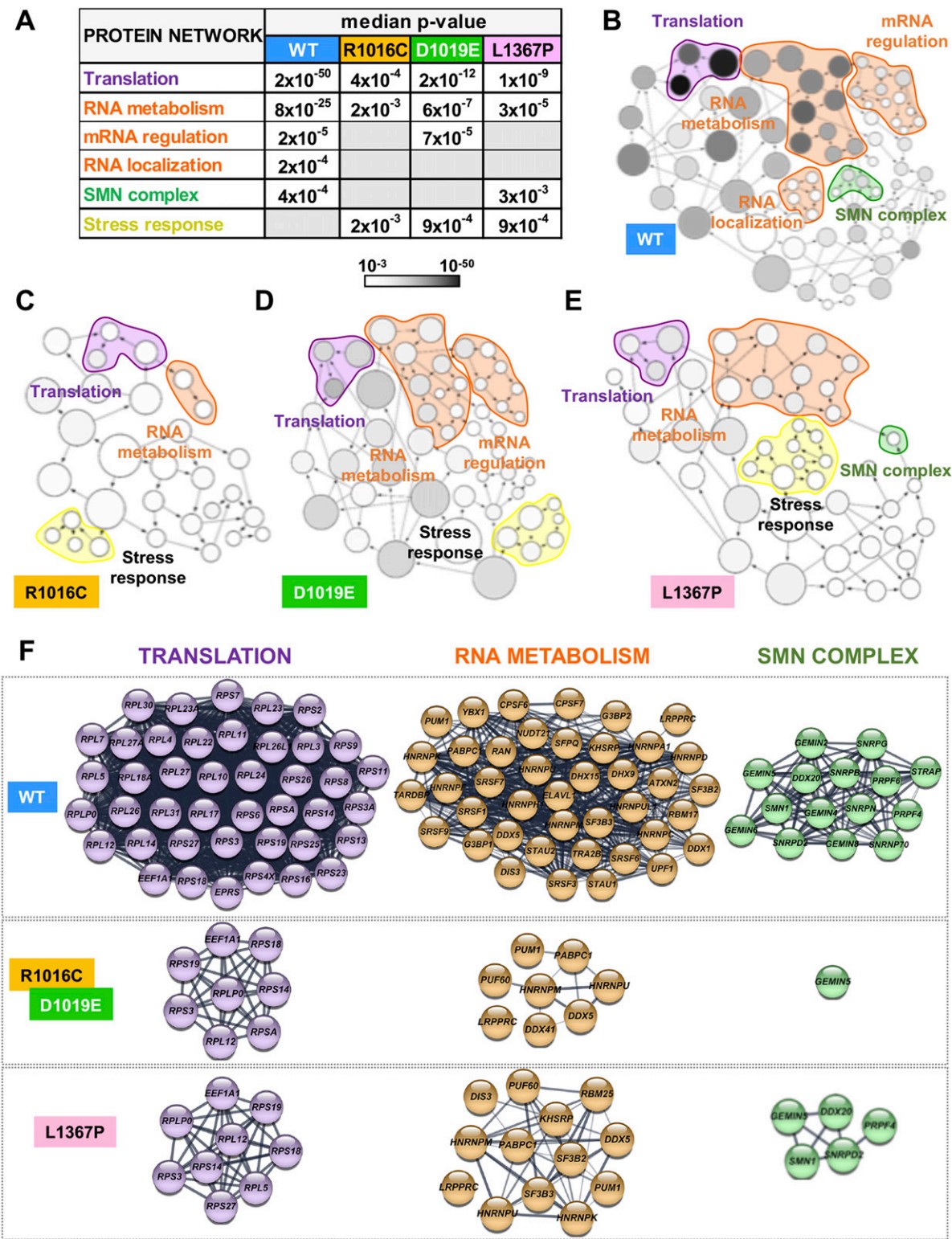

**A**

| PROTEIN NETWORK | median p-value | | | |
|---|---|---|---|---|
| | WT | R1016C | D1019E | L1367P |
| Translation | $2\times10^{-50}$ | $4\times10^{-4}$ | $2\times10^{-12}$ | $1\times10^{-9}$ |
| RNA metabolism | $8\times10^{-25}$ | $2\times10^{-3}$ | $6\times10^{-7}$ | $3\times10^{-5}$ |
| mRNA regulation | $2\times10^{-5}$ | | $7\times10^{-5}$ | |
| RNA localization | $2\times10^{-4}$ | | | |
| SMN complex | $4\times10^{-4}$ | | | $3\times10^{-3}$ |
| Stress response | | $2\times10^{-3}$ | $9\times10^{-4}$ | $9\times10^{-4}$ |

**Figure 3. Functional networks of proteins associated with G5$_{845-1508}$ variants.**
**(A)** Statistical difference (median *P*-values) obtained for protein networks associated to each of the G5$_{845-1508}$ variant obtained with the application BiNGO (Cytoscape platform). The *P*-values of the networks obtained for each G5$_{845-1508}$ variant relative to a complete human proteome are indicated. Empty lanes depict networks with *P*-values > 5 × 10$^{-3}$. **(B, C, D, E)** Networks obtained for the G5$_{845-1508}$ variants. The area of a node is proportional to the number of proteins in the test set annotated to the corresponding GO category, and the color intensity indicates the statistical significance of the node according to the gray scale bar. Arrows indicate branched nodes. Networks are shadowed violet, orange, green, or yellow according to the functional process. **(F)** STRING protein–protein networks associated to G5$_{845-1508}$-TAP. The cellular factors of TPR mutants correspond with the overlap between R1016C and D1019E samples.

of Gemin5 protein. Therefore, we sought to investigate the impact of the pathogenic variants on protein stability.

For this, each variant was inserted in the Xpress-G5$_{845-1508}$ construct and expressed in HEK293 cells. Twenty-four hours post-transfection, cells were treated with cycloheximide (CHX) for 16 h, and the level of G5$_{845-1508}$ protein accumulated in the presence or absence of CHX treatment was followed by immunoblotting (Fig 4A). No significant differences in the steady-state RNA levels of the WT and the mutants were observed by RTqPCR at the time of cell harvesting (Fig 4B). Quantification of the protein levels 12 and 16 h after CHX treatment relative to time 0 indicated that the relative intensity of the L1367P variant showed a strong decrease over time for G5$_{845-1508}$ protein (Fig 4C).

These results prompted us to investigate whether these differences could be also observed in the stability of the Gemin5 full-length protein (Fig S4A). No significant differences in the steady-state RNA levels were observed by RTqPCR (Fig S4B). Cells treated with CHX for 16 h showed a moderate protein decay for Gemin5 WT, R1016C, and D1019E (average protein intensity 0.47, 0.50, and 0.38, taking 1 as the value observed at time 0). In contrast, L1367P displayed a stronger decrease of Gemin5 protein levels 12 h (0.05) and 16 h (0.02) posttreatment (Fig S4C). Therefore, the defect in protein stability conferred by the L1367P variant is observed in both cases, presumably related to the reduced protein interactome.

## Mutations in the dimerization domain negatively affect ribosome association

We have reported earlier that Gemin5 behaves as a ribosome-binding protein (Francisco-Velilla et al, 2016) by using a G5$_{1-1287}$ construct that includes the WD40 repeat domains and the TPR-like dimerization domain (Fig 1A). Given that G5$_{845-1508}$-WT co-purified with a large number of the ribosomal proteins (Fig 3F), we asked whether this fragment interacts with the ribosome. Thus, we monitored the ribosome binding capacity using purified components, 80S ribosomal particles and His$_6$-tagged G5$_{845-1508}$ protein (Fig 5A, left panel). The results of the binding reaction were followed by immunoblotting using anti-His for G5$_{845-1508}$ and anti-P0 for the ribosome (Fig 5A, right panel). As a control, a binding reaction conducted with beads and the ribosomes yielded a rather weak signal with anti-P0 and no signal with anti-His (Fig 5A, lane 1), indicating that beads alone barely bind ribosomes. In contrast, the reaction conducted with His-G5$_{845-1508}$ and 80S ribosomes was positive for both components (Fig 5A, lane 2). The purified ribosomes alone (Fig 5A, lane 3) were negative with the anti-His antibody and positive for anti-P0. We conclude that G5$_{845-1508}$ can interact directly with the ribosome in the absence of other cellular components.

We focused our attention on ribosomal proteins belonging to the 60S and 40S subunits identified by mass spectrometry (Fig 3F). Glutathione S-transferase (GST) fusions of several ribosomal proteins were purified from bacteria and used in a pull-down assay with purified His-G5$_{845-1508}$. GST alone, used as a negative control of the pull-down, did not interact with His-G5$_{845-1508}$ (Fig 5B). Similar results were observed with L5. In contrast, GST fusions of the ribosomal proteins L3, L4, P0, S3A, S9, and S26 showed a direct interaction with His-G5$_{845-1508}$ in vitro. These results show that the G5$_{845-1508}$ fragment can interact with various ribosomal proteins belonging to the large and small ribosomal subunits, reinforcing the conclusion that there is a direct link between Gemin5 and the ribosome.

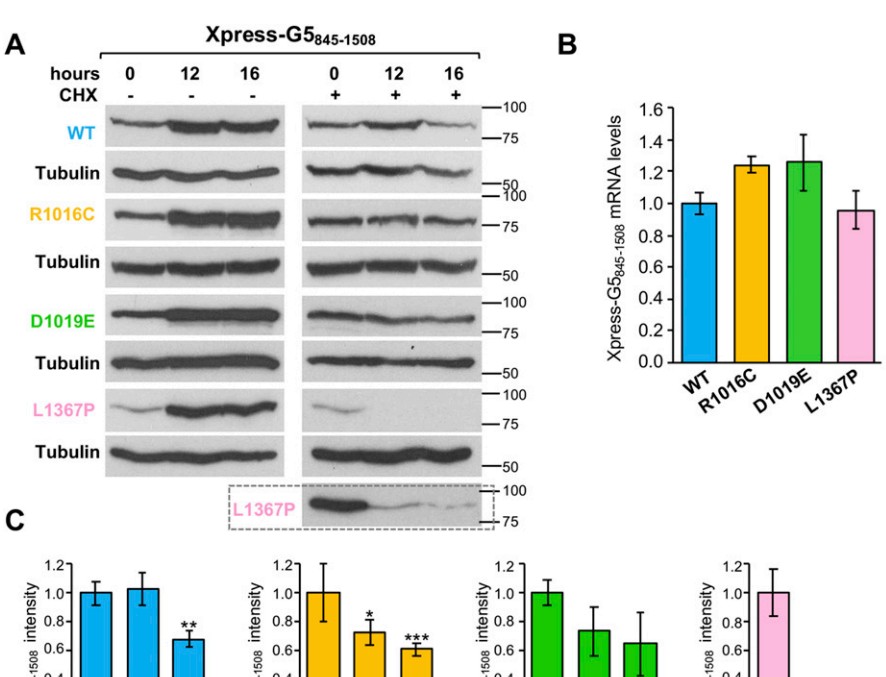

**Figure 4. Protein stability of G5$_{845-1508}$ carrying mutations R1016C, D1019E, or L1367P.**

**(A)** HEK293 cells expressing the wild-type version of G5$_{845-1508}$, side by side to the variants R1016C, D1019E, or L1367P during 12 h were treated (+) or not (−) with cycloheximide (CHX) for additional 16 h. Samples were taken at 0, 12, and 16 h post–CHX treatment. The intensity of each protein at the indicated time was determined by WB. A long exposure is shown for the L1367P protein. **(B)** Steady-state analysis of Xpress-G5$_{845-1508}$ mRNA levels present in transfected cells at the time of harvesting determined by RTqPCR. **(C)** Values represent the protein intensity (mean ± SEM) of three independent experiments relatively to time 0. Asterisks denote P-values (*P < 0.05, **P < 0.01, ***P < 0.001).

Source data are available for this figure.

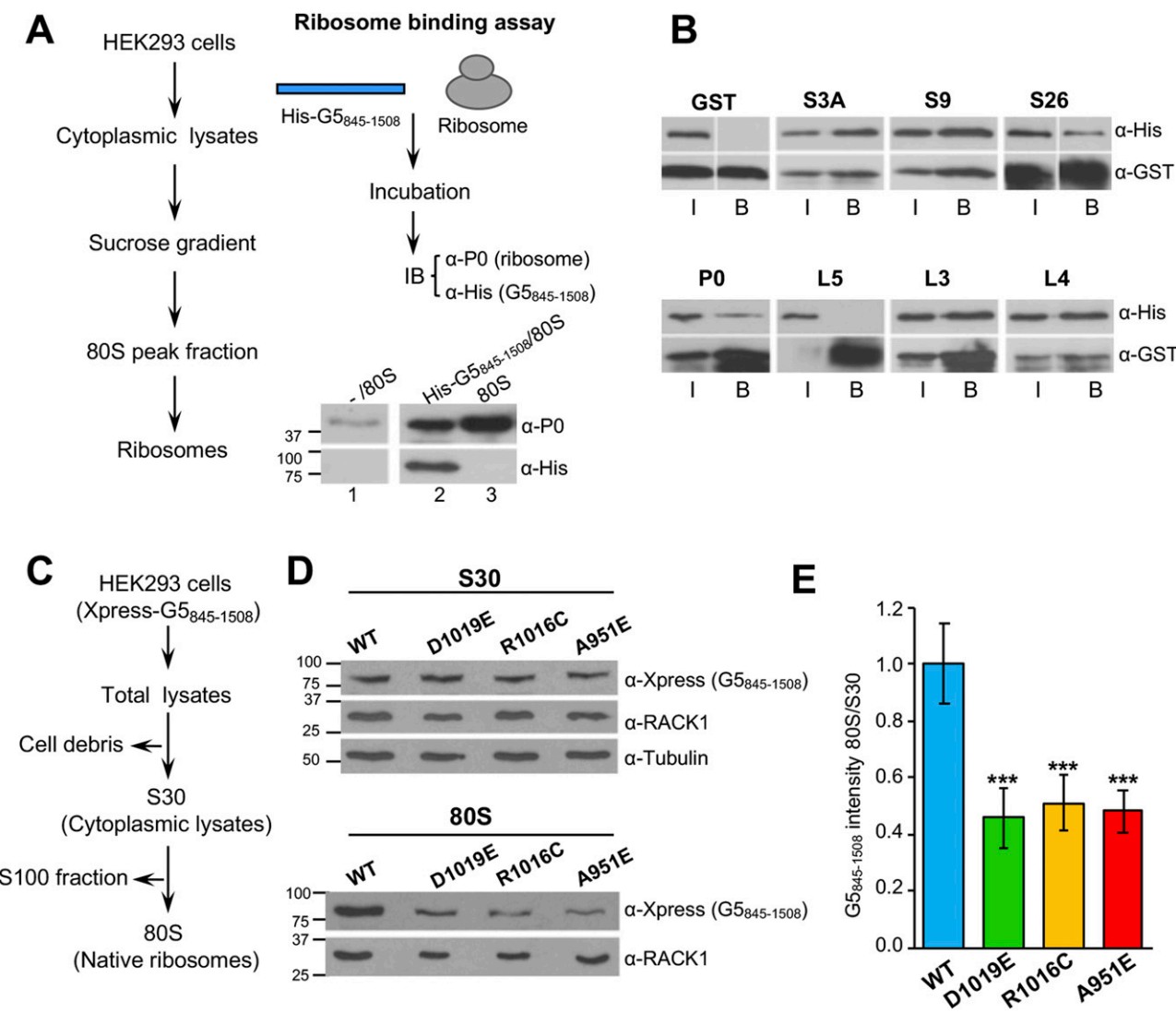

**Figure 5. Variants within the dimerization module abrogate the ribosome sedimentation capacity of G5₈₄₅₋₁₅₀₈.**

**(A)** Purified $G5_{845-1508}$ binds directly to the ribosome. Overview of the procedure used to isolate ribosomes 80S (left). Overview of the ribosome-binding assay. 80S ribosomes were incubated with beads alone (lane 1) or with Ni-agarose beads–bound His-$G5_{845-1508}$ (lane 2). After extensive washing, His-$G5_{845-1508}$ was immunodetected using the anti-His antibody. The antibody recognizing P0 was used to determine the presence of ribosomes bound to $G5_{845-1508}$ (lanes 1, 2, 3). **(B)** GST pull-down of $G5_{845-1508}$ with ribosomal proteins. GST protein alone was used as a negative control. $G5_{845-1508}$ was immunodetected using anti-His, and the GST fusion proteins were detected using anti-GST. I denotes input; B, Binding. **(C)** Overview of the procedure used to isolate native ribosomes. **(D)** Protein levels present in the S30 fraction (top) of HEK293 cells expressing Xpress-tagged $G5_{845-1508}$ proteins and sedimentation of these variants with native ribosomes (bottom). $G5_{845-1508}$ proteins were detected using anti-Xpress, the ribosomes with anti-RACK1. Tubulin was used as a loading control in S30 fractions. **(E)** Ratio of $G5_{845-1508}$ intensity in native ribosomes (80S) relative to the intensity observed in cytoplasmic lysates (S30). In all cases, values represent the mean ± SEM and asterisks denote P-values (***P < 0.001). Source data are available for this figure.

The severe decrease noticed in the Translation network of dimerization mutants (Fig 3F) prompted us to study whether these variants would impact ribosome binding. To this end, we analyzed the presence of $G5_{845-1508}$ in subcellular fractions enriched in native ribosomes. HEK293 cells expressing Xpress-tagged $G5_{845-1508}$-WT, -R1016C, -D1019E, and -A951E were used to prepare cytoplasmic lysates (S30) and native ribosomes (80S) (Fig 5C). The dimerization defective mutant $G5_{845-1508}$-A951E was used as a control. The Xpress-tagged $G5_{845-1508}$ proteins were immunodetected in the S30 fraction (Fig 5D, upper panel). Confirming the composition of the subcellular fractions, the ribosomal protein RACK1 was

detected in the S30 and 80S fractions (Rabl et al, 2011) (Fig 5D). The intensity of RACK1 indicated similar amounts of ribosome loading in all cases. However, we noticed a significant decrease in the intensity of all the mutant proteins in native ribosomes compared with the WT protein (Fig 5E), in agreement with the reduction of ribosomal proteins detected by mass spectrometry (Fig 3F). Representation of the 80S/S30 ratio readily indicated reduced ribosome sedimentation of the TPR-like mutants (Fig 5E). Together, these results demonstrate that $G5_{845-1508}$ sediments with native ribosomes and substitutions within the TPR-like domain impair this association.

### Gemin5 clinical mutants impaired in dimerization impinge on translation

The functional networks most significantly altered in the mutants analyzed in this study were RNA metabolism, mRNA regulation, and RNA localization (Fig 3A). These groups of proteins are engaged in regulation of mRNA stability, splicing, translation, and localization, among other processes. Beyond ribosomal proteins and eIFs, the mutants analyzed in this study were impaired in the association with several RBPs involved in translation regulation (Fig S5A). Thus, given that the R1016C and D1019E mutants showed defects in ribosome binding and in the ability to interact with factors related to translation regulation, we set up a functional assay in HEK293 cells co-expressing Xpress-tagged $G5_{845-1508}$ proteins and a luciferase reporter (Fig 6A). The results revealed that expression of Xpress-$G5_{845-1508}$-WT enhanced cap-dependent translation of luciferase relative to control cells, as expected (Moreno-Morcillo et al, 2020). In contrast, similar levels of expression of the proteins R1016C and D1019E failed to do so. In all cases, no significant differences in the steady-state levels of luciferase RNA were observed by RTqPCR at the time of cell harvesting (Fig 6B). Thus, we conclude that the variants found in patients, which reduce the ribosome association of Gemin5, abrogate translation stimulation driven by $G5_{845-1508}$.

We then asked whether these variants could affect the translation efficiency of selective mRNAs. For this, we measured the levels of selected mRNAs previously shown associated to Gemin5 (Francisco-Velilla et al, 2018) in translationally active polysomal fractions. The presence of RACK1 in the fractions of the gradient confirmed a similar loading of ribosomes in the samples prepared from cells expressing the WT and the protein variants R1016C and D1019E (Fig 6C). The lower density fractions of the gradient show a similar amount of $G5_{845-1508}$ proteins. However, a moderate reduction of R1016C and D1019E in 80S and polysomal fractions was observed relative to the WT protein, reinforcing the idea of a lower capacity to associate with ribosomes (Fig 6C). Then, measurement of association with polysomes of selected mRNAs —the H12 region of *Gemin5*, the AGO2, and PCBP1 mRNAs— indicated that Gemin5 mRNA (H12) was significantly enriched in the polysomal fractions with $G5_{845-1508}$-WT sample relative to the Input, although this enrichment was not observed with the R1016C and D1019E variants (Fig 6D). In contrast, the AGO2 mRNA was enriched in polysomal fractions both in the WT and mutant samples relative to the control. However, no significant differences were noticed in the association with polysomes of the mRNA encoding PCBP1 and in the mRNAs encoding the splicing factor SF3B3 and the RNA helicase DHX15 (Fig S5B).

We conclude that the defective mutants on the dimerization domain of Gemin5 impair the binding to polysomes of a specific mRNA, further reinforcing the involvement of this multitasking protein in selective translation.

## Discussion

We show in this study that *Gemin5* pathogenic variants mapping in the protein dimerization module (TPR-like) and the noncanonical RNA-binding domain (RBS1) impair the function of this multitasking protein in central cellular processes, including ribosome binding,

translation regulation, and protein–protein association. Of note, the clinical disorders associated with these *Gemin5* variants differ from those linked to SMN protein dysfunction (Kour et al, 2021; Saida et al, 2021). Consequently, these results lead us to propose that variants causing Gemin5 failure result in protein malfunction and, hence, are at the basis of the disease.

The protein Gemin5, which is expressed in all human tissues (Kim et al, 2014; Uhlén et al, 2015), is emerging as a multifunctional factor linked to human disease. A recent study reported *Gemin5* biallelic variants among patients presenting neuro-developmental disorders (Kour et al, 2021). The presence of *Gemin5* variants clustered in conserved residues of the dimerization domain, which provides a platform for protein–protein/RNA interactions (Moreno-Morcillo et al, 2020), reinforces the biological relevance of the TPR-like module for Gemin5 function. Fully consistent with this hypothesis, *Gemin5* biallelic variants have been recently associated with cerebellar atrophy and spastic ataxia in several human patients (Saida et al, 2021; Rajan et al, 2022). The phenotypic differences observed among individuals carrying similar but nonidentical substitutions in Gemin5 protein remain to be understood as the number of patients affected by this novel disease increase. However, these data sum up the observation that a null KO mouse is embryonic lethal as it also happens in flies (Gates et al, 2004; Borg et al, 2015).

The properties unveiled in our study of Gemin5 shed new light on this still poorly characterized protein. Combined, the TPR-like module and the RBS1 domain integrate key roles of the $G5_{845-1508}$ fragment: Gemin5 dimerization and selective RNA-binding. In turn, this supports the notion that the C-terminal region of the protein has different functions to the N-terminus. We show here that point mutations found in each of these motifs are detrimental for Gemin5 function. First, mutations R1016C and D1019E within the TPR-like domain impair endogenous Gemin5 recruitment in living cells. Second, mutation L1367P, a helix breaker residue, in a predicted helix of RBS1 induces protein instability, reducing its ability to interact with cellular RBPs but partially retaining the interaction with the SMN complex (Fig 7).

The high sequence conservation of the TPR-like sequence strongly suggests that the dimerization module plays a fundamental role for the architecture and activity of Gemin5. R1016 and D1019 form part of the loop $α12$-13 that forms a protrusion at the surface of the canoe-shaped dimer (Fig 1C). The presence of two highly conserved Pro residues (P1017 and P1020) in this loop and the network of electrostatic interactions involving residues R1016 and D1019 suggest that the conformation of this region is important for the stabilization of the dimer. We show that in the competitive environment of living cells, the full-length Gemin5 is recruited by the $G5_{845-1508}$ protein, whereas mutations R1016C and D1019E hamper this interaction (Fig 2B and C). Furthermore, the TPR-like module bearing mutation D1019E shows increased flexibility and lower stability (Fig 1E and F), although we did not detect a similar effect for mutation R1016C under the in vitro conditions tested. These results could be explained by the difference in the position of these residues within the structure: although R1016 protrudes to the surface, D1019 forms part of the internal part of this loop presumably contributing to a higher extent to a rigid conformation than R1016 (Fig 1C). However, several evidences support the relevance of R1016 in Gemin5 function. The mutant R1016C in the individual of case 1 is found in compound heterozygosity with a 5′ splice variant in intron 4 that is predicted to produce a truncated protein, suggesting that the R1016C mutation

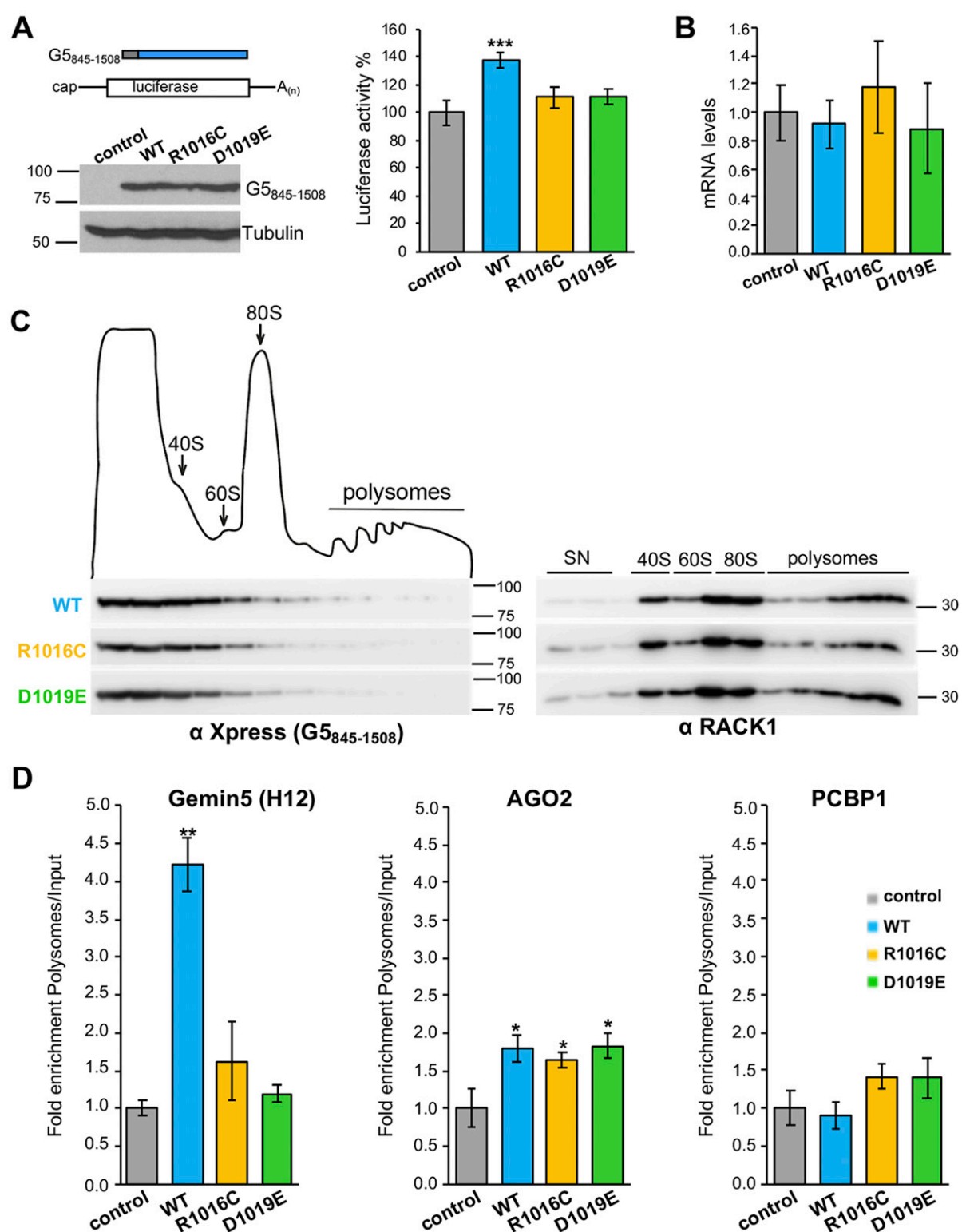

**Figure 6. Gemin5 mutants affecting dimerization fail to stimulate translation.**
**(A)** Diagram of the Xpress-G5$_{845-1508}$ proteins co-expressed with the luciferase reporter mRNA. Protein expression was monitored by WB using anti-Xpress, and tubulin was used as a loading control. Luciferase activity measured in HEK293 cell lysates expressing cap-mRNA co-transfected with Xpress-G5$_{845-1508}$-WT or the indicated variants. In all cases, luciferase values are normalized to cells expressing the empty vector conducted side by side. Values represent the mean ± SEM obtained in three independent assays. Asterisks denote *P*-values (***$P < 0.01$). **(B)** Steady-state mRNA levels analysis of the reporter were determined by RTqPCR in transfected cells at the time of harvesting for luciferase assays. **(C)** Polysome profiles prepared in 10–50% sucrose gradients loaded with total lysates of HEK293 cells expressing Xpress-G5$_{845-1508}$–

individually is crucial for the activity of Gemin5 in the cellular environment. In addition, the R1016C mutation has been detected in unrelated patients (Rajan et al, 2022), supporting an overall negative effect in protein function. Thus, the observed destabilizing effect of mutant D1019E in vitro, together with the negative effects of R1016C in living cells, strongly suggest that substitutions in the loop $\alpha$12-13 of the TPR-like moiety have a negative impact on protein dimerization. Hence, it is plausible that additional variants found in patients with similar disorders would also affect the Gemin5 dimerization properties.

Ribosome binding was initially reported for the extended N-terminal fragment (G5$_{1-1287}$) protein. This protein shares with G5$_{845-1508}$ the residues 845–1,287, including the entire dimerization domain. Considering that G5$_{845-1508}$ binds the ribosome using purified components (Fig 5A) and that this version of the protein sediments with native ribosomes (Fig 5C), we conclude that one of the key activities of Gemin5 depends on its interaction with the ribosome. Defects in the dimerization module impair the capacity to sediment with native ribosomes and also the association of G5$_{845-1508}$ with ribosomal proteins. In turn, these defects impinge on translation. We and others have shown that Gemin5 is involved in translation control (Pacheco et al, 2009; Workman et al, 2015; Francisco-Velilla et al, 2018; Garcia-Moreno et al, 2019). However, although G5$_{845-1508}$-WT enhances translation (Fig 6A), the mutants R1016C and D1019E fail to do so, in full agreement with the defective dimerization mutant A951E (Moreno-Morcillo et al, 2020).

Beyond protein stability and dimerization, the cellular processes negatively affected by the substitutions on the TPR-like and the RBS1 domains include protein–protein association (Fig 7). These processes are linked to RNA-driven pathways, involving the contribution of more than one functional domain of the Gemin5 protein. Further insight into the relative orientation and interconnection of the distinct functional domains awaits the structural resolution of the entire Gemin5 protein, which should also aid in predicting the damaging defect of new clinical variants.

In summary, our study shows that biallelic Gemin5 variants found in patients within the self-dimerization module and the noncanonical RNA-binding domain are linked to protein malfunction, reinforcing the critical role of Gemin5 in RNA-related pathways. The identification of molecular features associated with compound heterozygosity variants in Gemin5 gene opens new avenues for rational design of therapeutic strategies. In addition, these results provide evidence for the relevance of preserving balanced levels of Gemin5 protein to sustain gene expression regulation.

# Materials and Methods

## Constructs

The construct expressing G5$_{845-1508}$-WT (pcDNA3-Xpress-G5$_{845-1508}$) and Xpress-Gemin5-WT (pcDNA3-Xpress-G5) polypeptides were described (Fernandez-Chamorro et al, 2014; Francisco-Velilla et al, 2016). Likewise, the constructs pcDNA3-CTAP-G5$_{845-1508}$, pcDNA3-CTAP-G5$_{845-1508}$-A951E, pcDNA3-Xpress-G5$_{845-1508}$-A951E, and pOPINM_G5_TPR were reported (Moreno-Morcillo et al, 2020). Constructs pcDNA3-Xpress-G5-R1016C, pcDNA3-Xpress-G5-D1019E, pcDNA3-Xpress-G5-L1367P, pcDNA3-Xpress-G5$_{845-1508}$-R1016C, pcDNA3-Xpress-G5$_{845-1508}$-D1019E, pcDNA3-Xpress-G5$_{845-1508}$-L1367P, pcDNA3-CTAP-G5$_{845-1508}$-R1016C, pcDNA3-CTAP-G5$_{845-1508}$-D1019E, pcDNA3-CTAP-G5$_{845-1508}$-L1367P, pOPINM_G5_TPR-R1016C, and pOPINM_G5_TPR-D1019E were generated by QuickChange mutagenesis (Agilent Technologies) using specific primers (Table S1). pGEXKG plasmids expressing GST fusions of S3A, RACK1, L3, L4, L5, and P0 were previously described (Francisco-Velilla et al, 2016). Likewise, the constructs pGEXKG-S9 and pGEXKG-S26 were generated using standard procedures. See Table S1 for oligonucleotides sequences used for PCR. All plasmids were confirmed by DNA sequencing (Macrogen).

## Protein expression

The Gemin5 TPR-like domain was expressed and purified as previously reported (Moreno-Morcillo et al, 2020). Briefly, Escherichia coli BL21-Rosetta (DE3) pLysS cultures transformed with the plasmid pOPINM_G5_TPR (encoding the TPR-like domain fused to an N-terminal His6-MBP-tag cleavable by PreScission protease) were induced with 0.5 mM IPTG at 20°C overnight. The cells were harvested and resuspended in 40 ml of buffer A (20 mM Tris-HCl pH 8.0, 0.5 M NaCl, 10 mM imidazole, 5% glycerol, and 2 mM $\beta$-mercaptoethanol) supplemented with 0.5 mg/ml AEBSF [4-(2-aminoethyl) benzenesulfonyl fluoride hydrochloride] protease inhibitor. Following sonication, the clarified supernatant was loaded onto a 5-ml HisTrap HP column (Cytiva) equilibrated in buffer A, washed extensively with buffer A supplemented with 25 mM imidazole, and eluted by stepwise increase of imidazole to 250 mM. The protein was dialyzed overnight against buffer B (20 mM Tris-HCl pH 6.8, 50 mM NaCl, 5% glycerol, and 1 mM DTT) and GST-tagged PreScission protease (1/20$^{th}$ of total protein weight) was added into the dialysis bag to cleave the N-terminal tag. The sample was loaded onto a 5-ml HisTrap S HP column (Cytiva) equilibrated in buffer B, and the cleaved protein was eluted by increasing the salt concentration to 150 NaCl. The protein was concentrated by ultracentrifugation using an Amicon device with a 10 kD cutoff and loaded onto a size exclusion chromatography (SEC) HiLoad Superdex 200 16/60 column (Cytiva) equilibrated in buffer B. The sample eluted in a single peak and was concentrated as before. The sample was supplemented with 20% glycerol and flash-frozen in liquid nitrogen and stored at –80°C until use. Sample purity was evaluated by SDS–PAGE and Coomassie staining. Final protein purification yield was ~12 mg/l of cells. The R1016C and D1019E mutants were expressed and purified as the WT, except for increasing the NaCl concentration in the SEC buffer to 0.2 M to favor protein solubility. His-G5$_{845-1508}$ purified protein from S. cerevisiae was obtained as described (Fernandez-Chamorro et al, 2014).

WT, Xpress-G5$_{845-1508}$-R1016C, Xpress-G5$_{845-1508}$-D1019E, or the empty vector. The fractions of the gradient corresponding to 40S and 60S ribosomal subunits, 80S monosomes, and polysomes are indicated. Xpress-G5$_{845-1508}$ proteins and the ribosomal protein RACK1 were analyzed along the gradient fractions (20 $\mu$l) by WB using specific antibodies. **(D)** Histograms representing the fold enrichment in polysomes relative to the input samples of selected mRNAs (Gemin5 (region H12), AGO2, and PCBP1). Source data are available for this figure.

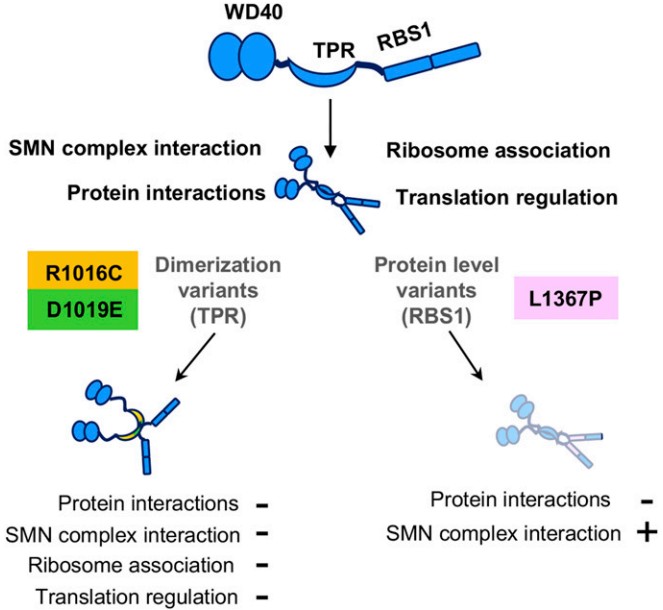

**Figure 7. Functions affected by pathogenic variants on the dimerization module and the RBS1 domain of Gemin5.**

Schematic representation of the Gemin5 protein with its functional domains: WD40, TPR-like dimerization module, and the noncanonical RBS1 domain. The spatial disposition of the domains is not known. Gemin5 dimerization variants show deficiencies in the interaction with members of the SMN complex, ribosome association, and translation stimulation. In contrast, the Gemin5 RBS1 variant retains these properties, but it is less stable (shady drawing denotes a reduced protein level). All variants show a reduced protein interactome. The combination of Gemin5 variants in patients and a dimerization defective mutant (R1016C or D1019E) with a protein level deficiency (L1367P) results in protein malfunction associated to disease.

## SEC coupled to multi-angle light scattering (SEC-MALS)

A total of 250 $\mu$l of purified TPR-like domain at 1–3 mg/ml were fractionated by SEC using a Superdex 200 10/300 column (Cytiva) equilibrated in 20 mM Tris-HCl pH 6.8, 0.2 M NaCl, 5% glycerol, and 1 mM DTT and an ÄKTA purifier (Cytiva) at a flow rate of 0.5 ml/min. The eluted samples were characterized by measuring the refractive index and multi-angle light scattering (MALS), using Optilab T-rEX and DAWN 8⁺ (Wyatt). Data were analyzed using the Astra6 software (Wyatt) to obtain the molar mass of each protein and represented using GraphPad Prism 9.

## Trypsin limited proteolysis

A total of 4 $\mu$g of total protein in 20 $\mu$l of buffer B were mixed with 5 $\mu$l of trypsin at 0.02–0.32 $\mu$g/ml and incubated at 22°C for 30 min. The reaction was stopped by adding SDS–PAGE sample loading buffer, and the samples were analyzed on 15% SDS–PAGE. The 30 kD band corresponding to the noncleaved protein band was quantified with program ImageJ.

## Thermal stability assays (Thermofluor)

Protein stability was measured by differential scanning fluorimetry (Niesen et al, 2007) using a 7500 Real-Time PCR System (Applied

Biosystems; Thermo Fisher Scientific) and a MicroAmp optical 96-well reaction plate sealed with film. The samples, prepared in a final volume of 20 $\mu$l, contained 2.5 $\mu$M TPR-like protein in buffer 20 mM Tris–HCl pH 6.8, 0.2 M NaCl, 5% glycerol and 1 mM DTT, and 5x SYPRO orange (Thermo Fisher Scientific). Fluorescence changes were monitored from 20°C to 85°C with increments of 1°C/min. Data were processed and analyzed with program GraphPad Prism 9.

## Protein complexes isolation by TAP

HEK293 human cells were cultured in Falcon© six-well plates with DMEM supplemented with 5% (vol/vol) FBS. Cells at 80% of confluency were transfected using Lipofectamine LTX (Thermo Fisher Scientific) and were harvested 24 hpt. The complexes associated to the TAP-tagged proteins were purified as described (Francisco-Velilla et al, 2016). Briefly, the extract from the TEV protease digestion of the first IgG Sepharose (Cytiva) purification was subsequently subjected to a second calmodulin (Agilent Technologies) purification step. Purified proteins were precipitated with 10% trichloroacetic acid at 4°C overnight, pelleted at 14,000 g for 15 min at 4°C, washed three times with 1 ml of acetone, and dissolved in SDS–loading buffer. An aliquot (20%) was analyzed on silver-stained SDS–PAGE gels to visualize the purification of proteins associated to G5$_{845-1508}$-TAP polypeptides.

## In-gel digestion and mass spectrometry analysis

Two independent biological replicates of TAP samples obtained for G5$_{845-1508}$-R1016C-TAP, G5$_{845-1508}$-D1019E-TAP, and G5$_{845-1508}$-L1367P-TAP were applied onto a 10% SDS–PAGE gel. The protein bands concentrated in the stacking/resolving gel interface were visualized by Coomassie staining. The gel pieces were destained in acetonitrile:water (ACN:H$_2$O, 1:1), were reduced and alkylated, and then digested in situ with sequencing grade trypsin (Promega) (Shevchenko et al, 1996). The gel pieces were dried and re-swollen in 50 mM ammonium bicarbonate, pH 8.8, with 60 ng/$\mu$l trypsin at 5:1 protein:trypsin (w/w) ratio. The tubes were kept in ice for 2 h and incubated at 37°C for 12 h. Digestion was stopped by the addition of 1% TFA. The desalted protein digest was dried, resuspended in 10 $\mu$l of 0.1% formic acid, and analyzed by RP-LC–MS/MS in an Easy-nLC II system coupled to an ion trap LTQ-Orbitrap-Velos-Pro hybrid mass spectrometer (Thermo Fisher Scientific). The peptides were concentrated (on-line) by reverse-phase chromatography using a 0.1 × 20 mm C18 RP precolumn (Thermo Fisher Scientific) and then separated using a 0.075 × 250 mm C18 RP column (Thermo Fisher Scientific) operating at 0.3 $\mu$l/min. Peptides were eluted using a 120-min dual gradient from 5 to 25% solvent B in 90 min followed by gradient from 25 to 40% solvent B over 120 min (Solvent A: 0.1% formic acid in water, solvent B: 0.1% formic acid, 80% acetonitrile in water). ESI ionization was done using a Nano-bore emitters Stainless Steel ID 30 $\mu$m (Proxeon) interface. The Orbitrap resolution was set at 30,000. Peptides were detected in survey scans from 400 to 1,600 amu (1 $\mu$scan), followed by 20 data-dependent MS/MS scans (Top 20), using an isolation width of 2 u (in mass-to-charge ratio units), normalized collision energy of 35%, and dynamic exclusion applied during 30 s periods.

Peptide identification from raw data was carried out using the PEAKS Studio X (Zhang et al, 2012) search engine (Bioinformatics Solutions Inc.). Database search was performed against UniProt-*Homo sapiens* FASTA (decoy-fusion database). The following constraints were used for the searches: tryptic cleavage after Arg and Lys, up to two missed cleavage sites, and tolerances of 20 ppm for precursor ions and 0.6 D for MS/MS fragment ions, and the searches were performed allowing optional Met oxidation and Cys carbamidomethylation. False discovery rates for peptide spectrum matches were limited to 0.01. Only those proteins with at least two distinct peptides being discovered from LC/MS/MS analyses were considered reliably identified (Supplemental Data 1).

The mass spectrometry proteomics data have been deposited to the ProteomeXchange Consortium with the dataset identifier PXD028959 and PXD028959.

### BiNGO and STRING analysis

The Biological Networks Gene Ontology application (BiNGO) was used to assess the overrepresentation of proteins associated with $G5_{845-1508}$-TAP variants and to determine the statistical significance of overrepresented proteins relative to a complete human proteome (Maere et al, 2005). Nodes overrepresented on the proteins associating with these variants relative to a whole human proteome belong to functional networks. The results were visualized on the Cytoscape platform (Shannon et al, 2003). The biological processes nodes were classified according to a hypergeometric test in the default mode, false discovery rate <0.05. *P*-values for the over-represented nodes were used to compute the average statistical significance of the network.

STRING software was used to depict the physical and functional interactions among the factors belonging to translation, RNA metabolism, and SMN complex networks (https://string-db.org).

### Protein stability assays

HEK293 human cells were transfected with the pcDNA3-Xpress-$G5_{845-1508}$ or pcDNA3-Xpress-G5 constructs. For cycloheximide (CHX) chase experiments, CHX (100 $\mu$g/ml) (Merck) was added to stop translation at 12- or 24-h post-transfection (hpt) for Xpress-$G5_{845-1508}$ or Xpress-Gemin5 proteins, respectively. Cells were harvested immediately (time 0) and 12 and 16 h post-CHX treatment. Cell lysates were prepared in 100 $\mu$l lysis buffer (50 mM Tris–HCl, pH 7.8, 100 mM NaCl, and NP40 0.5%), and the total protein concentration was determined by Bradford assay.

### RNA quantification

To measure the mRNA steady-state levels, total RNA was isolated from lysates prepared from cells harvested 24 hpt, expressing the corresponding plasmid as described (Francisco-Velilla et al, 2018). On the other hand, the total RNA from cytoplasmic lysates (Inputs) and polysome-bound RNA from the polysomal fraction 9 was extracted using TRIzol (Thermo Fisher Scientific), isopropanol precipitated, and resuspended in RNase-free $H_2O$. Reverse-transcriptase (RT) reaction was performed to synthesize cDNA

from equal amounts of the purified total RNA samples using SuperScript III (Thermo Fisher Scientific) and hexanucleotide mix (Merck) as primer. For quantitative PCR (qPCR), the pair of oligonucleotides 5′Luciferase/3′Luciferase (Francisco-Velilla et al, 2018) were used. The pairs of primers Xpress-s/Xpress-as, AGO2-s/AGO2-as, PCBP1-s/PCBP1-as, SF3B3-s/SF3B3-as, and DHX15-s/DHX15-as (Table S1) were designed (Primer3 software, http://bioinfo.ut.ee/primer3-0.4.0/primer3/) and tested for amplification efficiency. qPCR was carried out using the NZYSupreme qPCR Green Master Mix (NZytech) according to the manufacturer's instructions on an CFX-384 Fast Realtime PCR system (Bio-Rad). Values were normalized against the constitutive MYO5A RNA (Francisco-Velilla et al, 2018). The comparative cycle threshold method (Schmittgen & Livak, 2008) was used to quantify the results.

### Purification of 80S ribosomes

Fractions corresponding to 80S peak from polysome profiles loaded with HEK293 cell lysates were collected and pooled at 48,000 rpm using a T865 rotor 2 h, 4°C. The 80S pellet was resuspended in 10 mM HEPES pH 7, 10 mM $MgCl_2$, 50 mM KCl, and 5 mM $\beta$-ME and stored at −70°C. Ribosome concentration was calculated as 1 A260 unit = 20 pmol/ml 80S ribosome (Chen et al, 2014).

### Ribosome binding assay

His-tagged $G5_{845-1508}$ (His-$G5_{845-1508}$) (4 pmol) were coupled to Ni-agarose resin (25 $\mu$l of beads suspension) (QIAGEN) during 1 h, at 4°C in binding buffer (RBB) (50 mM TrisOAc, pH 7.7, 50 mM KOAc, 5 mM Mg $(OAc)_2$, 10 mM DTT, and 30 $\mu$g/ml tRNA). Unbound protein was removed by three washes with RBB, spinning at 14,000$g$ 3 min at 4°C. Beads–protein complexes, resuspended in 100 $\mu$l of RBB, were incubated with 80S ribosomes (0.7 pmol) during 1 h, at 4°C. After three washes of the beads complexes with RBB supplemented with NP40 0.05%, spinning at 14,000$g$ 3 min at 4°C, beads-bound proteins were dissolved in SDS-loading buffer, heated at 92°C 3 min, resolved by SDS–PAGE, and detected by WB.

### GST pull-down assay

The ribosomal proteins of interest were prepared as GST fusions as described (Francisco-Velilla et al, 2015). For binding, the GST fusion protein (4 $\mu$g) bound to the glutathione resin (Cytiva) was incubated with the His-$G5_{845-1508}$ purified protein, in five volumes of binding buffer (50 mM HEPES pH 7.4, 100 mM NaCl, 2 mM DTT, 2 mM $MgCl_2$, 0.5% Igepal CA-630, and 10% glycerol) 2 h, at 4°C in a rotating wheel. Beads were pelleted at 3,000 g 2 min at 4°C and washed three times with five volumes of binding buffer, rotating the reaction tube 5 min at 4°C. Finally, the beads were boiled in SDS-loading buffer, and proteins were resolved by SDS–PAGE and were detected by WB.

### Subcellular fractionation (S30 and native ribosomes)

HEK293 cells grown to 70–80% confluence in 2 P100 dishes were transfected with Xpress-$G5_{845-1508}$ constructs. Twenty-four hours later, cells were washed with ice-cold PBS and lysed in buffer 1 (15 mM Tris–HCl pH 7.4, 80 mM KCl, 5 mM $MgCl_2$, 1% Triton-X-100, and

protease inhibitors [Complete mini; Roche]). Cell debris was discarded by spinning at 14,000$g$ 10 min 4°C. The supernatant (S30 fraction) was ultracentrifuged at 95,000 rpm during 1 h 30 min using the TLA100.2 rotor yielded the S100 fraction (supernatant) and the native ribosomes (ribosomes plus associated factors). The pellets corresponding to native ribosomes were resuspended in 100 $\mu$l of buffer 1. The total protein content in S30 fractions was measured by the Bradford assay; the ribosome concentration was determined as 14 units $A_{260}$ = 1 mg/ml.

### Luciferase activity assays

HEK293 cells were cotransfected with plasmids expressing luciferase in a cap-dependent way (pCAP-luc) (Lozano et al, 2018) and the Xpress-G5$_{845–1508}$ constructs or the empty vector. Cell lysates were prepared 24 h post-transfection in 100 $\mu$l lysis buffer. The protein concentration in the lysate was determined by Bradford assay. Luciferase activity (RLU)/$\mu$g of total protein was internally normalized to the value obtained with the empty vector performed side by side.

### Polysome profiles, RNA isolation, and analysis

Polysome profiles were prepared from HEK293 cells (about 1 × 10$^7$ per gradient) transfected with the constructs expressing the Xpress-G5$_{845–1508}$ proteins as described (Francisco-Velilla et al, 2016). Briefly, cells were washed with ice-cold PBS containing 100 $\mu$g/ml cycloheximide to block ribosomes in the elongation step. Then, cells were lysed with buffer A (15 mM Tris–HCl pH 7.4, 80 mM KCl, 5 mM MgCl$_2$, and 100 $\mu$g/ml cycloheximide), supplemented with 1% (vol/vol) Triton X-100, 40 U/ml RNaseOUT (Thermo Fisher Scientific), and protease inhibitors (Complete mini; Roche). Cytoplasmic lysates obtained by centrifugation at 14,000$g$ 10 min at 4°C were loaded into a linear 10–50% (wt/vol) sucrose gradient in buffer A and centrifuged at 39,000 rpm in a SW40 Ti rotor 2 h 15 min at 4°C. Gradients were fractionated by upward displacement with 87% (vol/vol) glycerol using an density-gradient fractionator, monitoring A260 continuously (ISCO UA-5 UV monitor). Fractions (12 fractions of 1 ml) were collected from gradients.

### Immunodetection

The immunodetection of G5$_{845-1508}$ proteins were performed with different antibodies depending on the protein tag (anti-CBP [Abcam] for G5$_{845-1508}$-TAP, anti-Xpress [Thermo Fisher Scientific] for Xpress-G5$_{845–1508}$, or anti-His [Merck] for His-G5$_{845–1508}$). The anti-Gemin5 (Novus) antibody was used to detect the endogenous Gemin5 protein. The endogenous proteins tubulin, RACK1, and P0 were immunodetected with anti-Tubulin (Merck), anti-RACK1 (Santa Cruz), and 3BH5 (anti-P0) (Vilella et al, 1991) antibodies. GST fusion proteins were detected with the anti-GST antibody (Santa Cruz).

### Statistical analyses

Statistical analyses for experimental data were performed as follows. Each experiment was repeated independently at least three times. Values represent the mean ± SEM. We computed $P$-values for a difference in distribution between two samples with the unpaired two-tailed $t$ test. Differences were considered significant when $P <$ 0.05. The resulting $P$-values were graphically illustrated in figures with asterisks as described in figure legends.

## Data Availability

The mass spectrometry proteomics data have been deposited to the ProteomeXchange Consortium via the PRIDE (Perez-Riverol et al, 2019) partner repository with the dataset identifier PXD028959 and PXD028959. All data generated or analyzed during this study are included in the manuscript and supporting files.

## Supplementary Information

## Acknowledgments

We thank the Centro de Biologia Molecular Severo Ochoa (CBMSO) Proteomics Unit for help with the LC–MS/MS analysis, J Ramajo for technical assistance, MA Rodriguez for sharing the density-gradient fractionator, and C Gutierrez for valuable comments on the manuscript. We also thank the families and Doctors for sharing with us the clinical and genetic information of patients. This work was supported by the Spanish Ministerio de Ciencia e Innovación (MICIN) and Fondo Europeo de Desarrollo Regional (AEI/FEDER UE) grants BFU2017-84492-R, PID2020-115096RB-I00 (to E Martinez-Salas), and RTI2018-098084-B-100 (to S Ramón-Maiques), Comunidad de Madrid (B2017/BMD-3770 to E Martinez-Salas), and an Institutional grant from Fundación Ramón Areces (to E Martinez-Salas).

### Author Contributions

R Francisco-Velilla: conceptualization, data curation, formal analysis, investigation, and writing—original draft.
A Embarc-Buh: data curation, formal analysis, and investigation.
F del Cano-Ochoa: formal analysis and investigation.
S Abellan: investigation.
M Vilar: formal analysis and investigation.
S Alvarez: data curation, formal analysis, and investigation.
A Fernandez-Jaen: conceptualization, data curation, formal analysis, and investigation.
S Kour: investigation.
D Rajan: data curation and investigation.
UB Pandey: conceptualization, formal analysis, supervision, investigation, and writing—review and editing.
S Ramon-Maiques: data curation, formal analysis, supervision, investigation, and writing—original draft.
E Martinez-Salas: conceptualization, data curation, formal analysis, supervision, funding acquisition, investigation, and writing—original draft, review, and editing.

## Conflict of Interest Statement

The authors declare that they have no conflict of interest.

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
