## [Reviewer comments · Life Science Alliance]

Life Science Alliance

Functional and structural deficiencies of Gemin5 variants associated with neurological disorders

Rosario Francisco-Velilla, Azman Embarc-Buh, Francisco del Caño-Ochoa, Salvador Abellan, Marçal Vilar, Sara Alvarez, Alberto Fernandez-Jaen, Sukhleen Kour, Deepa Rajan, Udai Pandey, Santiago Ramon-Maiques, and Encarnacion Martinez-Salas

DOI: <https://doi.org/10.26508/lsa.202201403>

Corresponding author(s): Encarnacion Martinez-Salas, Centro de Biologia Molecular Severo Ochoa, CSIC-UAM

Review Timeline:

Submission Date:	2022-02-10
Editorial Decision:	2022-03-07
Revision Received:	2022-03-14
Editorial Decision:	2022-03-20
Revision Received:	2022-03-21
Accepted:	2022-03-22

Scientific Editor: Novella Guidi

Transaction Report:

March 7, 2022

Re: Life Science Alliance manuscript #LSA-2022-01403-T

Prof. Encarnacion Martinez-Salas
Centro de Biología Molecular
Genome Dynamics and Function
Nicolas Cabrera, 1, Cantoblanco
Madrid 28049
Spain

Dear Dr. Martinez-Salas,

Thank you for submitting your manuscript entitled "Functional and structural deficiencies of Gemin5 variants associated with neurological disorders" to Life Science Alliance. The manuscript was assessed by expert reviewers, whose comments are appended to this letter. We invite you to submit a revised manuscript addressing the Reviewer comments.

Thank you for this interesting contribution to Life Science Alliance. We are looking forward to receiving your revised manuscript.

Sincerely,

B. MANUSCRIPT ORGANIZATION AND FORMATTING:

Reviewer #1 (Comments to the Authors (Required)):

Francisco-Velilla et al. analyzed the effect of pathogenic mutations in Gemin5 associated with neurological disease. By using biophysical methods, they show that the mutants that disrupt dimerized TPR of Gemin5 leads to protein instability. Overall the manuscript is well organized and provides solid data to support its conclusion. I supports its publication in its current form.

Reviewer #2 (Comments to the Authors (Required)):

Francisco-Velilla et al. present very nice and solidly performed experiments that link mutations in Gemin 5 found in patients with neurodevelopmental disorders with Gemin 5 function. In recent years Gemin 5 emerged as an important RNA-binding protein that integrates multiple aspects of cellular RNA biology, yet its possible role in human pathophysiology remained largely unappreciated. In this manuscript, the authors describe Gemin 5 mutations found in two patient families and conduct experiments to link these mutations to Gemin 5 cellular functions. The experiments are well designed and executed, and the data strongly supports the conclusions of the manuscript. The following suggestions are meant to clarify the manuscript and to improve the strengths of the conclusions.

1. The first sentence in the results sections indicates that 3 patients were sequenced, but the subsequent text and data only refer to two cases, and Table 1 refers to only one patient. Some clarification is needed.
2. What are the endogenous levels of Gemin 5 in patient cells? Having this information would considerably strengthen the notion that these mutations (in particular L1367P) are impacting protein stability
3. Discussion. Could the authors elaborate on whether the differences in patient phenotypes correlate with the loss of interacting proteins with various mutants? In other words, does the data provide any hint on why the patient's phenotypes are different?
4. It is quite interesting that the dimerization defective A951 mutant is also very unstable, but the other mutants within the TPR domain are not. Only L1367P which is in the RBS1 domain. Could the authors provide some possible explanation or the hypothesis of why this might be the case?

Reviewer #3 (Comments to the Authors (Required)):

In this work, Francisco-Velilla et al have investigated the impact of Gemin5 missense variants found in compound heterozygosity in patients developing neurodevelopmental disorders. Gemin 5 is part of the SMN complex and has historically been associated with the delivery of snRNAs to snRNPs but also to various other activities in the cytoplasm, that are mainly involved in translation regulation. The variants that have been analyzed in this study were found in patients with compound heterozygosity and include a splice variant in intron 4 of the Gemin 5 gene and three missense mutations in conserved positions: R1016C, D1019E and L1367P.

Overall, this is a straightforward work where the authors have introduced these mutations in their respective functional domains within Gemin5 and investigated various physicochemical properties such as melting temperatures, flexibility, protein stability, and dimerization. The work was then complemented by an assessment of the impact of these mutants on the binding to other factors involved in the translation and RNA metabolism factors.

A few additions and clarifications should nonetheless be considered by the authors:

- 1) In Figure 2, the dimerization data are convincing and show that the R1016C and D1019E variants can affect dimerization in an intermediate manner compared to the A951E (positive control) substitution and the L1367P variant (that does not affect this process. Considering these results, it would have been therefore interesting if the authors had also performed a IHC analysis of the expressed proteins in HEK293 cells. Do the A951E/L1367P mutants occupy a different nuclear/cytosolic localization compared to R1016C/D1016E?.
- 2) Considering the results in Figures 4C and EV4 have the authors considered that the D1019E mutation could actually be stabilizing protein stability?. After all, in Figure 4 the amount Xpress G5 845-1508 fragment does not change during the 0-16

hour time course and in EV4 the Xpress Gemin5 with this mutation seems to decrease less than WT.

RESPONSE to REVIEWERS

We are very grateful to the reviewers for their thoughtful comments and suggestions that have helped us improve the quality of our current manuscript. All three reviewers acknowledged the significance of our work revealing the failures of GEMIN5 pathogenic variants. We believe that we have appropriately addressed the issues raised by the reviewers. We are providing a point-to-point response to the comments below.

Reviewer #1 (Comments to the Authors (Required)):

Francisco-Velilla et al. analyzed the effect of pathogenic mutations in Gemin5 associated with neurological disease. By using biophysic methods, they show that the mutants that disrupt dimerized TPR of Gemin5 leads to protein instability. Overall the manuscript is well organized and provides solid data to support its conclusion. I supports its publication in its current form.

RESPONSE: We thank the reviewer for commenting on the importance and significance of our current work using biophysical methods to study the dimerization properties of Gemin5. We appreciate the kind words on the organization of our manuscript.

Reviewer #2 (Comments to the Authors (Required)):

Francisco-Velilla et al. present very nice and solidly performed experiments that link mutations in Gemin 5 found in patients with neurodevelopmental disorders with Gemin 5 function. In recent years Gemin 5 emerged as an important RNA-binding protein that integrates multiple aspects of cellular RNA biology, yet its possible role in human pathophysiology remained largely unappreciated. In this manuscript, the authors describe Gemin 5 mutations found in two patient families and conduct experiments to link these mutations to Gemin 5 cellular functions. The experiments are well designed and executed, and the data strongly supports the conclusions of the manuscript. The following suggestions are meant to clarify the manuscript and to improve the strengths of the conclusions.

We thank the reviewer for the kind comments and also for giving us an opportunity to revise our manuscript to clarify the conclusions.

1. The first sentence in the results sections indicates that 3 patients were sequenced, but the subsequent text and data only refer to two cases, and Table 1 refers to only one patient. Some clarification is needed.

RESPONSE: We thank the reviewer for raising this point. To address this issue we have modified Table 1 to include the data from the 3 individuals. Also, we have modified the text (p. 4) to read: "Index case 1 is one individual with a splicing variant on intron 4 in one allele (frequency 60%) that is predicted to yield a truncated protein and a missense variant in exon 22 in the other allele (frequency 50%) producing the R1016C substitution within the TPR-like domain (Table 1) (Fig 1A). This individual is a female with intellectual disability, autism disorder, delay statural-ponderal growth, microcephaly, and mild dysmorphic features, but no motor problems. Both parents and a sister with monoallelic *Gemin5* variants are healthy, reinforcing the crucial impact of compound heterozygosity in disease. A second case corresponds to a family where two siblings (patients 2 and 3 in revised Table 1) with severe to mild neurological disorders carry a D1019E missense substitution within the TPR-like moiety in one allele and a L1367P substitution within the RBS1 region in the other allele (Fig 1B). Individuals 2 and 3 are female and male, respectively, with ataxia, hypotonia, developmental delay, cerebellar atrophy, motor delay, and cognitive delay (Kour et al., 2021)."

Table 1. Patients whole exome sequencing^{1,2,3}

Chr.	Gene	Intron/exon	Nucleotide change	Amino acid change	Mutation type
5	Gemin5	4	NM_015465.4: c.662-2A>G	-	5' splice site
5	Gemin5	22	NM_015465.4: c.3046C>T	p.Arg1016Cys	missense
5	Gemin5	22	NM_015465.5: c.3057C>A	p.Asp1019Glu	missense
5	Gemin5	26	NM_015465.5: c.41000T>C	p.L1367P	missense
5	Gemin5	22	NM_015465.5: c.3057C>A	p.Asp1019Glu	missense
5	Gemin5	26	NM_015465.5: c.41000T>C	p.L1367P	missense

¹ Female, compound heterozygosity, intellectual disability, autism disorder, delay staturo-ponderal growth, microcephaly, mild dysmorphic features, no motor problems.

² Female, compound heterozygosity, ataxia, hypotonia, developmental delay, severe cerebellar atrophy, motor delay, mild cognitive delay, slowed visual pursuit and mild oculomotor apraxia (Kour et al., 2021).

³ Male, compound heterozygosity, ataxia, hypotonia, moderate cerebellar atrophy, motor delay, tremor, hyperreflexia and mild dysarthria (Kour et al., 2021).

2. What are the endogenous levels of Gemin 5 in patient cells? Having this information would considerably strengthen the notion that these mutations (in particular L1367P) are impacting protein stability

RESPONSE: Thank you for this comment. Addressing this point requires fresh sample from the patients. We show below a Western blot image obtained with blood samples from patient #1 (proband) and her healthy sister (control). We isolated white blood cells from peripheral blood sample (PBLs) prior to lyse cells and load equal amounts of protein in PAGE. Gemin5 was detected using anti-Gemin5 (Novus). We do not detect major differences in the intensity of the protein in both samples. For patients #2 and #3, which include the L1367P variant, we do not have access to samples. We therefore believe that the data of individual #1 will not add significant information to the manuscript. However, we may include this in supplementary material if requested.

3. Discussion. Could the authors elaborate on whether the differences in patient phenotypes correlate with the loss of interacting proteins with various mutants? In other words, does the data provide any hint on why the patient's phenotypes are different?

RESPONSE: This is a highly relevant issue which needs to be studied in the future, given that the number of patients clinically identified with defects in Gemin5 gene are very limited. However, as the reviewer suggested, we have included a brief sentence on the Discussion to comment on the phenotypic differences reported in patients: "Fully consistent with this

hypothesis, *Gemin5* biallelic variants have been recently associated with cerebellar atrophy and spastic ataxia in several human patients (Saida et al., 2021; Rajan et al., in press). The phenotypic differences observed among individuals carrying similar but non-identical substitutions in *Gemin5* remain to be understood as the number of patients affected by this novel disease increase. However, these data sum up the observation that a null KO mouse is embryonic lethal, as it also happens in flies (Gates et al., 2004; Borg et al., 2015).

4. It is quite interesting that the dimerization defective A951 mutant is also very unstable, but the other mutants within the TPR domain are not. Only L1367P which is in the RBS1 domain. Could the authors provide some possible explanation or the hypothesis of why this might be the case?

RESPONSE: We can only hypothesize on the reasons for protein instability of L1367P. As already pointed out in the manuscript (p. 5): "L1367 locates within a predicted helix of RBS1, with substitutions to Val, Phe, and Cys, but not to Pro that is a helix-breaker amino acid. Therefore, each of these clinical variants affect conserved residues of the protein which is likely to affect the physiological functions". In support of this possibility, other *Gemin5* mutants analyzed in the manuscript by Kour et al (2021) carrying substitution to Proline are less stable than the WT protein. A related sentence is also present in the Discussion: "Second, mutation L1367P, a helix breaker residue, in a predicted helix of RBS1 induces protein instability, reducing its ability to interact with cellular RBPs but partially retaining the interaction with the SMN complex (Fig 7)."

Reviewer #3 (Comments to the Authors (Required)):

In this work, Francisco-Velilla et al have investigated the impact of *Gemin5* missense variants found in compound heterozygosity in patients developing neurodevelopmental disorders. *Gemin 5* is part of the SMN complex and has historically been associated with the delivery of snRNAs to snRNPs but also to various other activities in the cytoplasm, that are mainly involved in translation regulation. The variants that have been analyzed in this study were found in patients with compound heterozygosity and include a splice variant in intron 4 of the *Gemin 5* gene and three missense mutations in conserved positions: R1016C, D1019E and L1367P.

Overall, this is a straightforward work where the authors have introduced these mutations in their respective functional domains within *Gemin5* and investigated various physicochemical properties such as melting temperatures, flexibility, protein stability, and dimerization. The work was then complemented by an assessment of the impact of these mutants on the binding to other factors involved in the translation and RNA metabolism factors.

We are grateful the reviewer for going through our paper in-depth and giving his/her constructive feedback that allowed us to improve the presentation of work.

A few additions and clarifications should nonetheless be considered by the authors:

1) In Figure 2, the dimerization data are convincing and show that the R1016C and D1019E variants can affect dimerization in an intermediate manner compared to the A951E (positive control) substitution and the L1367P variant (that does not affect this process. Considering these results, it would have been therefore interesting if the authors had also performed a IHC analysis of the expressed proteins in HEK293 cells. Do the A951E/L1367P mutants occupy a different nuclear/cytosolic localization compared to R1016C/D1016E?.

RESPONSE: This an excellent point which will be addressed in future studies. Gemin5 is rather abundant mainly cytoplasmic protein, although there are also detectable amounts of protein in the nucleus. Therefore, this type of immunohistochemistry (IHC) studies requires in depth quantification of protein levels in different cell compartments by complementary biochemical and functional approaches which are beyond the scope of this article.

2) Considering the results in Figures 4C and EV4 have the authors considered that the D1019E mutation could actually be stabilizing protein stability?. After all, in Figure 4 the amount Xpress G5 845-1508 fragment does not change during the 0-16 hour time course and in EV4 the Xpress Gemin5 with this mutation seems to decrease less than WT.

RESPONSE: We thank the reviewer for raising this point. The results shown for the protein stability of D1019E mutant in Fig 4 and EV4 have the same trend. However, because of the differences in the error bars, those shown in Fig 4 (Xpress-p85) are not significantly different from time 0, while those in fig EV4 are statistically different. We do not appreciate major differences among them as to infer protein stability differences.

March 20, 2022

RE: Life Science Alliance Manuscript #LSA-2022-01403-TR

Prof. Encarnacion Martinez-Salas
Centro de Biología Molecular Severo Ochoa, CSIC-UAM
Genome Dynamics and Function
Nicolas Cabrera, 1, Cantoblanco
Madrid 28049
Spain

Dear Dr. Martinez-Salas,

Thank you for submitting your revised manuscript entitled "Functional and structural deficiencies of Gemin5 variants associated with neurological disorders". We would be happy to publish your paper in Life Science Alliance pending final revisions necessary to meet our formatting guidelines.

- please add the twitter handle of your host institute/organization as well as your own or/and one of the authors in our system
- please make sure that author names in the manuscript and our system match
- please rename your EV figures and tables as supplementary figures and tables; please also update the callouts in the text (e.g. Fig EV2A would be Fig S2A)
- please add a callout for Figure 6D in your main manuscript text

A. FINAL FILES:

B. MANUSCRIPT ORGANIZATION AND FORMATTING:

Sincerely,

Reviewer #2 (Comments to the Authors (Required)):

I am satisfied with the revisions and clarifications provided by the authors, and support publications of this revised manuscript without further changes.

Reviewer #3 (Comments to the Authors (Required)):

Authors have answered well to all comments from this reviewer

March 22, 2022

RE: Life Science Alliance Manuscript #LSA-2022-01403-TRR

Prof. Encarnacion Martinez-Salas
Centro de Biología Molecular Severo Ochoa, CSIC-UAM
Genome Dynamics and Function
Nicolas Cabrera, 1, Cantoblanco
Madrid 28049
Spain

Dear Dr. Martinez-Salas,

Thank you for submitting your Research Article entitled "Functional and structural deficiencies of Gemin5 variants associated with neurological disorders". It is a pleasure to let you know that your manuscript is now accepted for publication in Life Science Alliance. Congratulations on this interesting work.

DISTRIBUTION OF MATERIALS:

Again, congratulations on a very nice paper. I hope you found the review process to be constructive and are pleased with how the manuscript was handled editorially. We look forward to future exciting submissions from your lab.

Sincerely,
